# Partial Fourier loss for semantic segmentation: experiments in the spectral domain

**Hoel Kervadec** [ID][1]                                                            H.T.G.KERVADEC@UVA.NL

[1] *Universiteit van Amsterdam (UvA), Instituut van Informatica (IvI), qurAI group*

**Editors:** Accepted for publication at MIDL 2025

**Keywords:** Semantic segmentation, Fourier transform, Partial supervision

## 1. Introduction

In the context of fully-supervised semantic segmentation, the object boundary is quite interesting: it is, at the same time, one of the most important "detail" to get right—if we want a prediction to be clicinally useful—but also, it is where most of the label uncertainty is located. Here, it is useful to make the distinction between the *ground-truth*, which is the ideal "truth" that we may never have access to, and a *label*, which is what we have to use for training deep neural networks. In a sense, we can consider that the label is the ground-truth plus annotator bias plus some noise.

In this short paper, we experiment with the Fourier transform as a loss for neural networks, in place of more common losses (cross-entropy, Dice, Boundary loss). We start from a simple assumption: the annotator bias and noise in the annotation is located in the high-frequencies of the annotation, whereas the low-frequency information contains the core of what we want to learn.

## 2. Formulation

Let us define, without loss of generality, a binary label as a function over the D-dimensional (discrete) image space $\Omega \subset \mathbb{N}^D$ as $y^{(\cdot)} : \Omega \to \{0,1\}$, and a predicted segmentation (more accurately, its predicted probabilities) as $s_\theta^{(\cdot)} : \Omega \to [0,1]$, where $\theta$ represents the learnable parameters of network $\mathcal{N}$. For some input $x \in \mathbb{R}^{M \times \Omega}$ with $M$ modalities, we have therefore $s_\theta \triangleq \mathcal{N}(x; \theta)$. $p = (\mathsf{x}, \mathsf{y}, ...) \in \Omega \subset \mathbb{N}^D$ is the spatial domain decomposed across its different axises, and $q = (\mathsf{u}, \mathsf{v}, ...) \in \Xi \subset \mathbb{N}^D$ is the same for the spectral domain. (Here the dots are used to express a continuation in case of 3-dimensional images or more.)

As a reminder (Gonzalez and Woods, 2018, Section 2.4), and updated to our notation, the Fourier transform of a segmentation $s$ (be it $y$ or $s_\theta$) is as follows and is easily reversed:

$$\mathcal{F}_s^{(\mathsf{u},\mathsf{v},...)} \triangleq \sum_{\mathsf{x}=0}^{\mathsf{X}-1} \sum_{\mathsf{y}=0}^{\mathsf{Y}-1} \sum_{...} s^{(\mathsf{x},\mathsf{y},...)} e^{-i2\pi\left(\frac{\mathsf{u}\mathsf{x}}{\mathsf{X}} + \frac{\mathsf{v}\mathsf{y}}{\mathsf{Y}} + ...\right)}, \tag{1}$$

$$s^{(\mathsf{x},\mathsf{y},...)} = \sum_{\mathsf{u}=0}^{\mathsf{U}-1} \sum_{\mathsf{v}=0}^{\mathsf{V}-1} \sum_{...} \mathcal{F}_s^{(\mathsf{u},\mathsf{v},...)} e^{-i2\pi\left(\frac{\mathsf{u}\mathsf{x}}{\mathsf{U}} + \frac{\mathsf{v}\mathsf{y}}{\mathsf{V}} + ...\right)}. \tag{2}$$

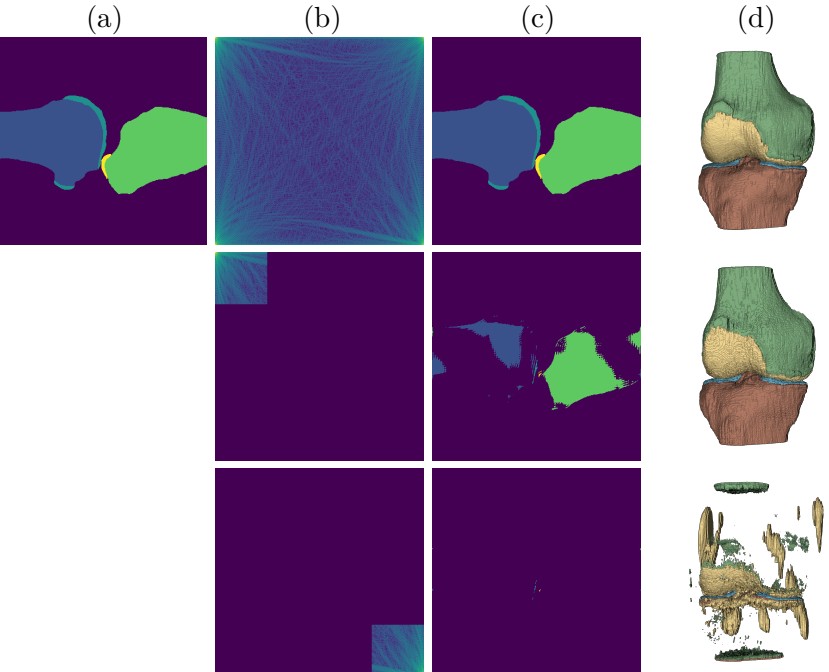

Figure 1: (a) Original label, (b) (Subset) of the frequency domain kept, (c) Segmentation reconstructed from (b), (d) test prediction when supervising only (b).

A "full" loss on the Fourier domain could look like this:

$$\mathcal{L}_{\mathcal{F}}(y, s_\theta) \triangleq \sum_{u=0}^{U-1} \sum_{v=0}^{V-1} \sum_{\cdots} \left( \mathcal{F}_y^{(u,v,\dots)} - \mathcal{F}_{s_\theta}^{(u,v,\dots)} \right)^2, \tag{3}$$

whereas a "partial" loss on the Fourier domain could be, when keeping only the low-frequencies or high-frequencies, respectively:

$$\mathcal{L}_{\mathcal{F}_L}(y, s_\theta) \triangleq \sum_{u=0}^{\frac{U}{L}-1} \sum_{v=0}^{\frac{V}{L}-1} \sum_{\cdots} \left( \mathcal{F}_y^{(u,v,\dots)} - \mathcal{F}_{s_\theta}^{(u,v,\dots)} \right)^2, \tag{4}$$

$$\mathcal{L}_{\mathcal{F}^H}(y, s_\theta) \triangleq \sum_{u=U-\frac{U}{H}}^{U-1} \sum_{v=V-\frac{V}{H}}^{V-1} \sum_{\cdots} \left( \mathcal{F}_y^{(u,v,\dots)} - \mathcal{F}_{s_\theta}^{(u,v,\dots)} \right)^2. \tag{5}$$

A higher $L$ or $H$ results in a smaller fraction ($\frac{1}{L^2}$ or $\frac{1}{H^2}$) of the spectral domain being used. For multiclass segmentation, we simply repeat the operation over each class.

## 3. Experiments

We perform experiments on two datasets: SegTHOR (Lambert et al., 2020) (CT of heart, esophagus, aorta and trachea, with a size of $512 \times 512 \times \sim 200$) and OAI (Almajalid et al.,

2022) (CT of knee bones and cartilages, $384 \times 384 \times 160$). Different neural network architectures were used (U-Net and not U-Net based), with no data augmentation for SegTHOR and with data augmentation for OAI. Adam optimizer was used for all experiments. No extra fine-tuning was performed for the learning rate or other hyper-parameters.

Despite some datasets having a single connected component per object, we did not use any post-processing. As such, to complement the Dice Score (DSC) as first metric, we resort to the Average Symmetric Surface Distance (ASSD), in millimeters (mm). To enable comparisons, we make the choice to define the ASSD between a label and an empty prediction as the diagonal of the scan. Experiments were performed on a single fold and we report only a single run: numbers should therefore not be considered as statistically significantly but merely indicative of "potential" performances.

Results are shown in Table 1, and we can see that when supervising only a fraction of the high-frequencies, performances quickly collapse, whereas the low-frequencies could be reduced much more.

Table 1: Test metrics for the two datasets and different loss variants.

| | SegTHOR | | OAI | |
|---|---|---|---|---|
| Loss | DSC (%) | ASSD (mm) | DSC (%) | ASSD (mm) |
| $\mathcal{L}_{\mathrm{CE}}$ | 76.3±16.1 | 3.230±2.070 | 91.5±07.6 | 0.229±0.098 |
| $\mathcal{L}_{\mathcal{F}}$ | 78.8±14.2 | 2.750±1.791 | 91.6±07.5 | 0.247±0.193 |
| $\mathcal{L}_{\mathcal{F}_2}$ | 76.0±14.7 | 3.016±1.887 | 91.9±07.2 | 0.220±0.092 |
| $\mathcal{L}_{\mathcal{F}_4}$ | 77.1±15.4 | 3.198±2.787 | 91.7±07.5 | 0.225±0.090 |
| $\mathcal{L}_{\mathcal{F}_8}$ | 75.5±16.1 | 2.804±1.772 | 90.9±08.2 | 0.237±0.095 |
| $\mathcal{L}_{\mathcal{F}_{16}}$ | 75.1±15.6 | 4.017±3.227 | 91.2±07.8 | 0.263±0.113 |
| $\mathcal{L}_{\mathcal{F}^2}$ | 76.1±14.7 | 2.849±1.478 | 91.8±07.3 | 0.221±0.096 |
| $\mathcal{L}_{\mathcal{F}^4}$ | 05.3±09.1 | 485.278±380.721 | 25.3±30.1 | 7.024±5.640 |

## 4. Conclusion

This short paper played with the Fourier transform as a loss for semantic segmentation. Unsurprisingly, supervising the whole Fourier domain is on par with other common losses (reporting only the cross-entropy here), and we will not dwell on this result.

What was surprising, however, is how little of the low-frequency domain was required to reach decent performance. This was a pleasant surprise to the author, and gives different insights and perspective on the *meaning* of supervision. Indeed, what is left of the spectral domain (as shown in Fig. 1) is not enough to reconstruct the segmentation itself, so it was not entirely expected[1] that it would be sufficient to train satisfactorily a segmentation neural network.

Resilience and generalization ability to noisy labels and annotator style is the next logical step to be studied. *Plausibly*, supervising only a fraction of the spectral domain may be a decent alternative to more complex schemes to handle uncertainty around the boundary.

---

1. Not entirely unexpected either, as $\mathcal{F}_s^{(0,\dots,0)} = \sum_{i \in \Omega} s^{(i)}$, which is the basis of (Kervadec et al., 2019) which supervised only the size of the object with some seed point as weak labels.

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
