# OpenReview forum: "Partial Fourier loss for semantic segmentation: experiments in the spectral domain"
_MIDL.io/2025/Short_Papers — MIDL 2025 - Short Papers_

### Official Review · Reviewer_P6QF · 2025-04-28

**Rating:** 4
**Confidence:** 4

**Summary:**

This paper proposes a loss function to train the segmentation network, which is applied to the Fourier domain. Based on the assumption that the annotator bias and noise in the labels are in high-frequency information, and the core semantic information is in the low-frequencies of the labels, the paper conducts experiments by varying the loss forms, which keep only the low-frequency or high-frequency information. The experimental results show that the whole Fourier domain information provides comparable performance with other common loss functions for the segmentation task. Also, the results report that a small fraction of high-/low-frequency degrade the segmentation performance.

**Strengths:**

- The paper presents a segmentation loss function that is applied to the Fourier domain of the segmentation mask, which can be adapted to various segmentation tasks.
- The paper reports how low-frequency or high-frequency information of the annotations is required to achieve high segmentation performance.

**Weaknesses:**

- The paper lacks descriptions of the network the authors used in the experiments.
- The performance of the proposed Fourier-domain segmentation loss function is only demonstrated on the knee bones.
- There is no description of why the whole Fourier domain loss function achieves higher performance than the cross-entropy loss function.

---

### Decision · Program_Chairs · 2025-05-01

Accept